# Dynamic Filter Networks

**Bert De Brabandere**[1]*
ESAT-PSI, KU Leuven, iMinds

**Xu Jia**[1]*
ESAT-PSI, KU Leuven, iMinds

**Tinne Tuytelaars**[1]
ESAT-PSI, KU Leuven, iMinds

**Luc Van Gool**[1,2]
ESAT-PSI, KU Leuven, iMinds
D-ITET, ETH Zurich
[1]`firstname.lastname@esat.kuleuven.be`   [2]`vangool@vision.ee.ethz.ch`

## Abstract

In a traditional convolutional layer, the learned filters stay fixed after training. In contrast, we introduce a new framework, the *Dynamic Filter Network*, where filters are generated dynamically conditioned on an input. We show that this architecture is a powerful one, with increased flexibility thanks to its adaptive nature, yet without an excessive increase in the number of model parameters. A wide variety of filtering operations can be learned this way, including local spatial transformations, but also others like selective (de)blurring or adaptive feature extraction. Moreover, multiple such layers can be combined, e.g. in a recurrent architecture.

We demonstrate the effectiveness of the dynamic filter network on the tasks of video and stereo prediction, and reach state-of-the-art performance on the moving MNIST dataset with a much smaller model. By visualizing the learned filters, we illustrate that the network has picked up flow information by only looking at unlabelled training data. This suggests that the network can be used to pretrain networks for various supervised tasks in an unsupervised way, like optical flow and depth estimation.

## 1   Introduction

Humans are good at predicting another view from related views. For example, humans can use their everyday experience to predict how the next frame in a video will differ; or after seeing a person's profile face have an idea of her frontal view. This capability is extremely useful to get early warnings about impinging dangers, to be prepared for necessary actions, etc. The vision community has realized that endowing machines with similar capabilities would be rewarding.

Several papers have already addressed the generation of an image conditioned on given image(s). Yim *et al.* [24] and Yang *et al.* [23] learn to rotate a given face to another pose. The authors of [16, 19, 18, 15, 12] train a deep neural network to predict subsequent video frames. Flynn *et al.* [3] use a deep network to interpolate between views separated by a wide baseline. Yet all these methods apply the exact same set of filtering operations on each and every input image. This seems suboptimal for the tasks at hand. For example, for video prediction, there are different motion patterns within different video clips. The main idea behind our work is to generate the future frames with parameters

adapted to the motion pattern within a particular video. Therefore, we propose a learnable parameter layer that provides custom parameters for different samples.

Our *dynamic filter module* consists of two parts: a *filter-generating network* and a *dynamic filtering layer* (see Figure 1). The filter-generating network dynamically generates sample-specific filter parameters conditioned on the network's input. Note that these are not fixed after training, like regular model parameters. The dynamic filtering layer then applies those sample-specific filters to the input. Both components of the dynamic filter module are differentiable with respect to the model parameters such that gradients can be backpropagated throughout the network. The filters can be convolutional, but other options are possible. In particular, we propose a special kind of dynamic filtering layer which we coin *dynamic local filtering layer*, which is not only sample-specific but also position-specific. The filters in that case vary from position to position and from sample to sample, allowing for more sophisticated operations on the input. Our framework can learn both spatial and photometric changes, as pixels are not simply displaced, but the filters possibly operate on entire neighbourhoods.

We demonstrate the effectiveness of the proposed dynamic filter module on several tasks, including video prediction and stereo prediction. We also show that, because the computed dynamic filters are explicitly calculated - can be visualised as an image similar to an optical flow or stereo map. Moreover, they are learned in a totally unsupervised way, i.e. without groundtruth maps.

The rest of paper is organised as follows. In section 2 we discuss related work. Section 3 describes the proposed method. We show the evaluation in section 4 and conclude the paper in section 5.

## 2   Related Work

**Deep learning architectures**   Several recent works explore the idea of introducing more flexibility into the network architecture. Jaderberg *et al.* [9] propose a module called Spatial Transformer, which allows the network to actively spatially transform feature maps conditioned on themselves without explicit supervision. They show this module is able to perform translation, scaling, rotation and other general warping transformations. They apply this module to a standard CNN network for classification, making it invariant to a set of spatial transformations. This seminal method only works with parametric transformations however, and applies a single transformation to the entire feature map(s). Patraucean *et al.* [15] extend the Spatial Transformer by modifying the grid generator such that it has one transformation for each position, instead of a single transformation for the entire image. They exploit this idea for the task of video frame prediction, applying the learned dense transformation map to the current frame to generate the next frame. Similarly, our method also applies a position specific transformation to the image or feature maps and takes video frame prediction as one testbed. In contrast to their work, our method generates the new image by applying dynamic local filters to the input image or feature maps instead of using a grid generator and sampler. Our method is not only able to learn how to displace a pixel, but how to construct it from an entire neighborhood, including its intensity (e.g. by learning a blur kernel).

In the context of visual question answering, Noh *et al.* [13] introduce a dynamic parameter layer which output is used as parameters of a fully connected layer. In that work, the dynamic parameter layer takes the information from another domain, *i.e.* question representation, as input. They further apply hashing to address the issue of predicting the large amount of weights needed for a fully connected layer. Different from their work, we propose to apply the dynamically generated filters to perform a filtering operation on an image, hence we do not have the same problem of predicting large amounts of parameters. Our work also shares similar ideas with early work on fast-weight networks [4], that is, having a network learn to generate context dependent weights for another network. However, we instantiate this idea as a convolution/local filtering operation with spatial information under consideration while they use a fully connected layer, and use it as an alternative for RNN. Most similar to our work, a dynamic convolution layer is proposed by Klein *et al.* [10] in the context of short range weather prediction and by Riegler *et al.* [17] for single image non-blind single image super resolution. Our work differs from theirs in that it is more general: dynamic filter networks are not limited to translation-invariant convolutions, but also allow position-specific filtering using a dynamic locally connected layer. Lastly, Finn *et al.* [2] recently independently proposed a mechanism called *(convolutional) dynamic neural advection* that is very similar to ours.

**New view synthesis** Our work is also related to works on new view synthesis, that is, generating a new view conditioned on the given views of a scene. One popular task in this category is to predict future video frames. Ranzato *et al.* [16] use an encoder-decoder framework in a way similar to language modeling. Srivastava *et al.* [19] propose a multilayer LSTM based autoencoder for both past frames reconstruction and future frames prediction. This work has been extended by Shi *et al.* [18] who propose to use convolutional LSTM to replace the fully connected LSTM in the network. The use of convolutional LSTM reduces the amount of model parameters and also exploits the local correlation in the image. Oh *et al.* [14] address the problem of predicting future frames conditioned on both previous frames and actions. They propose the encoding-transformation-decoding framework with either feedforward encoding or recurrent encoding to address this task. Mathieu *et al.* [12] manage to generate reasonably sharp frames by means of a multi-scale architecture, an adversarial training method, and an image gradient difference loss function. In a similar vein, Flynn *et al.* [3] apply a deep network to produce unseen views given neighboring views of a scene. Their network comes with a selection tower and a color tower, and is trained in an end-to-end fashion. This idea is further refined by Xie *et al.* [22] for 2D-to-3D conversion. None of these works adapt the filter operations of the network to the specific input sample, as we do, with the exception of [3, 22]. We'll discuss the relation between their selection tower and our dynamic filter layer in section 3.3.

**Shortcut connections** Our work also shares some similarity, through the use of shortcut connections, with the highway network [20] and the residual network [7, 8]. For a module in the highway network, the transform gate and the carry gate are defined to control the information flow across layers. Similarly, He *et al.* [7, 8] propose to reformulate layers as learning residual functions instead of learning unreferenced functions. Compared to the highway network, residual networks remove the gates in the highway network module and the path for input is always open throughout the network. In our network architecture, we also learn a referenced function. Yet, instead of applying addition to the input, we apply filtering to the input - see section 3.3 for more details.

## 3 Dynamic Filter Networks

In this section we describe our dynamic filter framework. A dynamic filter module consists of a filter-generating network that produces filters conditioned on an input, and a dynamic filtering layer that applies the generated filters to another input. Both components of the dynamic filter module are differentiable. The two inputs of the module can be either identical or different, depending on the task. The general architecture of this module is shown schematically in Figure 1. We explicitly model the transformation: invariance to change should not imply one becomes totally blind to it. Moreover, such explicit modeling allows unsupervised learning of transformation fields like optical flow or depth.

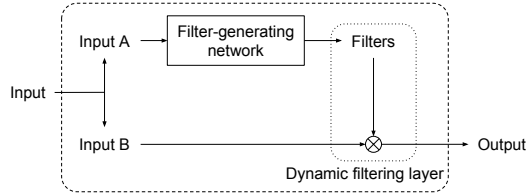

Figure 1: The general architecture of a Dynamic Filter Network.

For clarity, we make a distinction between *model parameters* and *dynamically generated parameters*. Model parameters denote the layer parameters that are initialized in advance and only updated during training. They are the same for all samples. Dynamically generated parameters are sample-specific, and are produced on-the-fly without a need for initialization. The filter-generating network outputs dynamically generated parameters, while its own parameters are part of the model parameters.

### 3.1 Filter-Generating Network

The filter-generating network takes an input $I_A \in \mathbb{R}^{h \times w \times c_A}$, where $h$, $w$ and $c_A$ are height, width and number of channels of the input $A$ respectively. It outputs filters $\mathcal{F}_\theta$ parameterized by parameters $\theta \in \mathbb{R}^{s \times s \times c_B \times n \times d}$, where $s$ is the filter size, $c_B$ the number of channels in input $B$ and $n$ the number of filters. $d$ is equal to 1 for dynamic convolution and $h \times w$ for dynamic local filtering, which we discuss below. The filters are applied to input $I_B \in \mathbb{R}^{h \times w \times c_B}$ to generate an output $G = \mathcal{F}_\theta(I_B)$, with $G \in \mathbb{R}^{h \times w \times n}$. The filter size $s$ determines the receptive field and is chosen depending on the

application. The size of the receptive field can also be increased by stacking multiple dynamic filter modules. This is for example useful in applications that may involve large local displacements.

The filter-generating network can be implemented with any differentiable architecture, such as a multilayer perceptron or a convolutional network. A convolutional network is particularly suitable when using images as input to the filter-generating network.

## 3.2 Dynamic Filtering Layer

The dynamic filtering layer takes images or feature maps $I_B$ as input and outputs the filtered result $G \in \mathbb{R}^{h \times w \times n}$. For simplicity, in the experiments we only consider a single feature map ($c_B = 1$) filtered with a single generated filter ($n = 1$), but this is not required in a general setting. The dynamic filtering layer can be instantiated as a dynamic convolutional layer or a dynamic local filtering layer.

**Dynamic convolutional layer.** A dynamic convolutional layer is similar to a traditional convolutional layer in that the same filter is applied at every position of the input $I_B$. But different from the traditional convolutional layer where filter weights are model parameters, in a dynamic convolutional layer the filter parameters $\theta$ are dynamically generated by a filter-generating network:

$$G(i, j) = \mathcal{F}_\theta(I_B(i, j)) \tag{1}$$

The filters are sample-specific and conditioned on the input of the filter-generating network. The dynamic convolutional layer is shown schematically in Figure 2(a). Given some prior knowledge about the application at hand, it is sometimes possible to facilitate training by constraining the generated convolutional filters in a certain way. For example, if the task is to produce a translated version of the input image $I_B$ where the translation is conditioned on another input $I_A$, the generated filter can be sent through a softmax layer to encourage elements to only have a few high magnitude elements. We can also make the filter separable: instead of a single square filter, generate separate horizontal and vertical filters that are applied to the image consecutively similar to what is done in [10].

**Dynamic local filtering layer.** An extension of the dynamic convolution layer that proves interesting, as we show in the experiments, is the dynamic local filtering layer. In this layer the filtering operation is not translation invariant anymore. Instead, different filters are applied to different positions of the input $I_B$ similarly to the traditional locally connected layer: for each position $(i, j)$ of the input $I_B$, a specific local filter $\mathcal{F}_\theta^{(i,j)}$ is applied to the region centered around $I_B(i, j)$:

$$G(i, j) = \mathcal{F}_\theta^{(\mathbf{i},\mathbf{j})}(I_B(i, j)) \tag{2}$$

The filters used in this layer are not only sample specific but also position specific. Note that dynamic convolution as discussed in the previous section is a special case of local dynamic filtering where the local filters are shared over the image's spatial dimensions. The dynamic local filtering layer is shown schematically in Figure 2b. If the generated filters are again constrained with a softmax function so that each filter only contains one non-zero element, then the dynamic local filtering layer replaces each element of the input $I_B$ by an element selected from a local neighbourhood around it. This offers a natural way to model local spatial deformations conditioned on another input $I_A$. The dynamic local filtering layer can perform not only a single transformation like the dynamic convolutional layer, but also position-specific transformations like local deformation. Before or after applying the dynamic local filtering operation we can add a dynamic pixel-wise bias to each element of the input $I_B$ to address situations like photometric changes. This dynamic bias can be produced by the same filter-generating network that generates the filters for the local filtering.

When inputs $I_A$ and $I_B$ are both images, a natural way to implement the filter-generating network is with a convolutional network. This way, the generated position-specific filters are conditioned on the local image region around their corresponding position in $I_A$. The receptive field of the convolutional network that generates the filters can be increased by using an encoder-decoder architecture. We can also apply a smoothness penalty to the output of the filter-generating network, so that neighboring filters are encouraged to apply the same transformation.

Another advantage of the dynamic local filtering layer over the traditional locally connected layer is that we do not need so many model parameters. The learned model is smaller and this is desirable in embedded system applications.

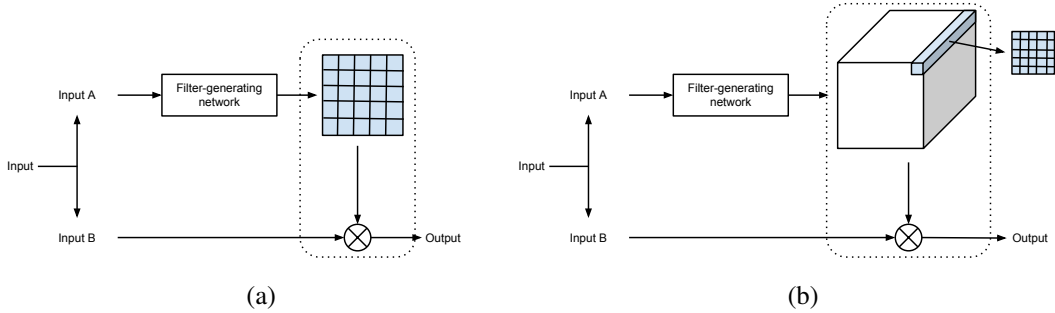

(a)                                                         (b)

Figure 2: *Left:* Dynamic convolution: the filter-generating network produces a single filter that is applied convolutionally on $I_B$. *Right:* Dynamic local filtering: each location is filtered with a location-specific dynamically generated filter.

## 3.3 Relationship with other networks

The generic formulation of our framework allows to draw parallels with other networks in the literature. Here we discuss the relation with the spatial transformer networks [9], the deep stereo network [3, 22], and the residual networks [7, 8].

**Spatial Transformer Networks**   The proposed dynamic filter network shares the same philosophy as the spatial transformer network proposed by [9], in that it applies a transformation conditioned on an input to a feature map. The spatial transformer network includes a localization network which takes a feature map as input, and it outputs the parameters of the desired spatial transformation. A grid generator and sampler are needed to apply the desired transformation to the feature map. This idea is similar to our dynamic filter network, which uses a filter-generating network to compute the parameters of the desired filters. The filters are applied on the feature map with a simple filtering operation that only consists of multiplication and summation operations.

A spatial transformer network is naturally suited for global transformations, even sophisticated ones such as a thin plate spline. The dynamic filter network is more suitable for local transformations, because of the limited receptive field of the generated filters, although this problem can be alleviated with larger filters, stacking multiple dynamic filter modules, and using multi-resolution extensions. A more fundamental difference is that the spatial transformer is only suited for spatial transformations, whereas the dynamic filter network can apply more general ones (e.g. photometric, filtering), as long as the transformation is implementable as a series of filtering operations. This is illustrated in the first experiment in the next section.

**Deep Stereo**   The deep stereo network of [3] can be seen as a specific instantiation of a dynamic filter network with a local filtering layer where inputs $I_A$ and $I_B$ denote the same image, only a horizontal filter is generated and softmax is applied to each dynamic filter. The effect of the selection tower used in their network is equivalent to the proposed dynamic local filtering layer. For the specific task of stereo prediction, they use a more complicated architecture for the filter-generating network.

**Residual Networks**   The core idea of ResNets [7, 8] is to learn a residual function with respect to the identity mapping, which is implemented as an additive shortcut connection. In the dynamic filter network, we also have two branches where one branch acts as a shortcut connection. This becomes

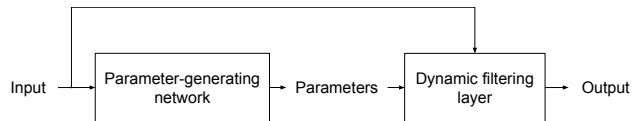

Figure 3: Relation with residual networks.

clear when we redraw the diagram (Figure 3). Instead of merging the branches with addition, we merge them with a dynamic filtering layer which is multiplicative in nature. Multiplicative interactions in neural networks have also been investigated by [21].

| Model | Moving MNIST | |
| --- | --- | --- |
| | # params | bce |
| FC-LSTM [19] | 142,667,776 | 341.2 |
| Conv-LSTM [18] | 7,585,296 | 367.1 |
| Spatio-temporal [15] | 1,035,067 | 179.8 |
| Baseline (ours) | 637,443 | 432.5 |
| **DFN (ours)** | 637,361 | 285.2 |

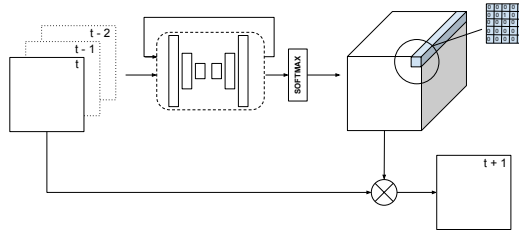

Table 1: *Left:* Quantitative results on Moving MNIST: number of model parameters and average binary cross-entropy (bce). *Right:* The dynamic filter network for video prediction.

# 4 Experiments

The Dynamic Filter Network can be used in different ways in a wide variety of applications. In this section we show its application in learning steerable filters, video prediction and stereo prediction. All code to reproduce the experiments is available at `https://github.com/dbbert/dfn`.

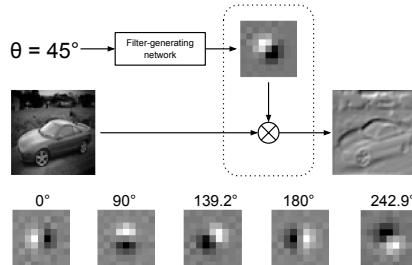

Figure 4: The dynamic filter network for learning steerable filters and several examples of learned filters.

## 4.1 Learning steerable filters

We first set up a simple experiment to illustrate the basics of the dynamic filter module with a dynamic convolution layer. The task is to filter an input image with a steerable filter of a given orientation $\theta$. The network must learn this transformation from looking at input-output pairs, consisting of randomly chosen input images and angles together with their corresponding output.

The task of the filter-generating network here is to transform an angle into a filter, which is then applied to the input image to generate the final output. We implement the filter-generating network as a few fully-connected layers with the last layer containing 81 neurons, corresponding to the elements of a 9x9 convolution filter. Figure 4 shows an example of the trained network. It has indeed learned the expected filters and applies the correct transformation to the image.

## 4.2 Video prediction

In video prediction, the task is to predict the sequence of future frames that follows the given sequence of input frames. To address this task we use a convolutional encoder-decoder as the filter-generating network where the encoder consists of several strided convolutional layers and the decoder consists of several unpooling layers and convolutional layers. The convolutional encoder-decoder is able to exploit the spatial correlation within a frame and generates feature maps that are of the same size as the frame. To exploit the temporal correlation between frames we add a recurrent connection inside the filter-generating network: we pass the previous hidden state through two convolutional layers and sum it with the output of the encoder to produce the new hidden state. During prediction, we propagate the prediction from the previous time step. Table 1 (right) shows a diagram of our architecture. Note that we use a very simple recurrent architecture rather than the more advanced LSTM as in [19, 18]. A softmax layer is applied to each generated filter such that each filter is encouraged to only have a few high magnitude elements. This helps the dynamic filtering layer to generate sharper images because each pixel in the output image comes from only a few pixels in the previous frame. To produce the prediction of the next frame, the generated filters are applied on the previous frame to transform it with the dynamic local filtering mechanism explained in Section 3.

**Moving MNIST** We first evaluate the method on the synthetic moving MNIST dataset [19]. Given a sequence of 10 frames with two moving digits as input, the goal is to predict the following 10 frames. We use the code provided by [19] to generate training samples on-the-fly, and use the provided test

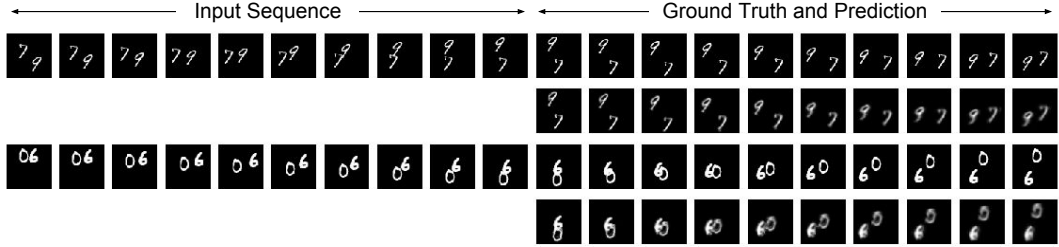

Figure 5: Qualitative results on moving MNIST. Note that the network has learned the bouncing dynamics and separation of overlapping digits. More examples and out-of-domain results are in the supplementary material.

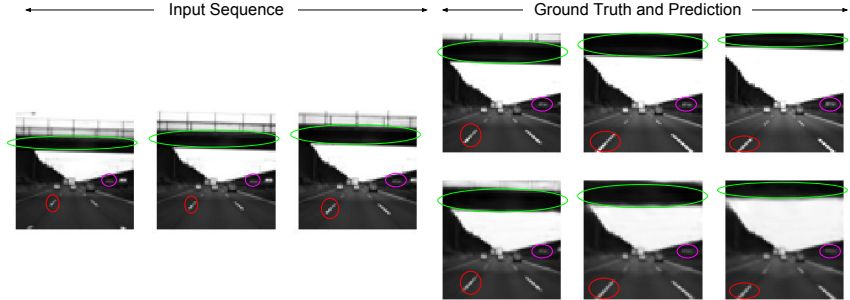

Figure 6: Qualitative results of video prediction on the Highway Driving dataset. Note the good prediction of the lanes (red), bridge (green) and a car moving in the opposite direction (purple).

set for comparison. Only simple pre-processing is done to convert pixel values into the range [0,1]. As the loss function we use average binary cross-entropy over the 10 frames. The size of the dynamic filters is set to 9x9. This allows the network to translate pixels over a distance of at most 4 pixels, which is sufficient for this dataset. Details on the hyper-parameter can be found in the available code.

We also compare our results with a **baseline** consisting of only the filter-generating network, followed by a $1 \times 1$ convolution layer. This way, the baseline network has approximately the same structure and number of parameters as the proposed dynamic filter network. The quantitative results are shown in Table 1 (left). Our method outperforms the baseline and [19, 18] with a much smaller model. Figure 5 shows some qualitative results. Our method is able to correctly learn the individual motions of digits. We observe that the predictions deteriorate over time, i.e. the digits become blurry. This is partly because of the model error: our model is not able to perfectly separate digits after an overlap, and these errors accumulate over time. Another cause of blurring comes from an artifact of the dataset: because of imperfect cropping, it is uncertain when exactly the digit will bounce and change its direction. The behavior is not perfectly deterministic. This uncertainty combined with the pixel-wise loss function encourages the model to "hedge its bets" when a digit reaches the boundary, causing a blurry result. This issue could be alleviated with the methods proposed in [5, 6, 11].

**Highway Driving**   We also evaluate our method on real-world data of a car driving on the highway. Compared to natural video like UCF101 used in [16, 12], the highway driving data is highly structured and much more predictable, making it a good testbed for video prediction. We add a small extension to the architecture: a dynamic per-pixel bias is added to the image before the filtering operation. This allows the network to handle illumination changes such as when the car drives through a tunnel.

Because the Highway Driving sequence is less deterministic than moving MNIST, we only predict the next 3 frames given an input sequence of 3 frames. We split the approximately $20,000$ frames of the 30-minute video into a training set of $16,000$ frames and a test set of $4,000$ frames. We train with a Euclidean loss function and obtain a loss of $13.54$ on the test set with a model consisting of $368,122$ parameters, beating the baseline which gets a loss of $15.97$ with $368,245$ parameters.

Figure 6 shows some qualitative results. Similar to the experiments on moving MNIST, the predictions get blurry over time. This can partly be attributed to the increasing uncertainty combined with an

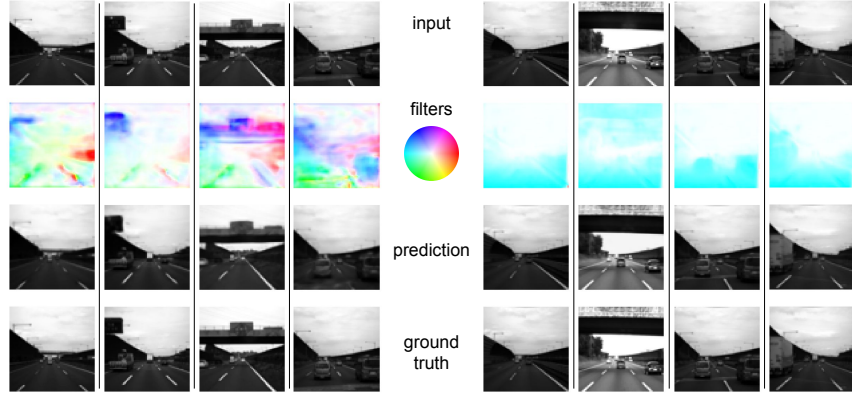

Figure 7: Some samples for video (*left*) and stereo (*right*) prediction and visualization of the dynamically generated filters. More examples and a video can be found in the supplementary material.

element-wise loss-function which encourages averaging out the possible predictions. Moreover, the errors accumulate over time and make the network operate in an out-of-domain regime.

We can visualize the dynamically generated filters of the trained model in a flow-like manner. The result is shown in Figure 7 and the visualization process is explained in the supplementary material. Note that the network seems to generate "valid" flow only insofar that it helps with minimizing its video prediction objective. This is sometimes noticeable in uniform, textureless regions of the image, where a valid optical flow is no prerequisite for correctly predicting the next frame. Although the flow map is not perfectly smooth, it is learned in a self-supervised way by only training on unlabeled video data. This is different from supervised methods like [1].

### 4.3 Stereo prediction

We define stereo prediction as predicting the right view given the left view of a stereo camera. This task is a variant of video prediction, where the goal is to predict a new view in space rather than in time, and from a single image rather than multiple ones. Flynn *et al.* [3] developed a network for new view synthesis from multiple views in unconstrained settings like musea, parks and streets. We limit ourselves to the more structured Highway Driving dataset and a classical two-view stereo setup.

We recycle the architecture from the previous section, and replace the square 9x9 filters with horizontal 13x1 filters. The network is trained and evaluated on the same train- and test split as in the previous section, with the left view as input and the right one as target. It reaches a loss of $0.52$ on the test set with a model consisting of $464,494$ parameters. The baseline obtains a loss of $1.68$ with $464,509$ parameters. The network has learned to shift objects to the left depending on their distance to the camera, as shown in Figure 7 (right). The results suggest that it is possible to use the proposed dynamic filter network architecture to pre-train networks for optical flow and disparity map estimation in a self-supervised manner using only unlabeled data.

## 5 Conclusion

In this paper we introduced Dynamic Filter Networks, a class of networks that applies dynamically generated filters to an image in a sample-specific way. We discussed two versions: dynamic convolution and dynamic local filtering. We validated our framework in the context of steerable filters, video prediction and stereo prediction. As future work, we plan to explore the potential of dynamic filter networks on other tasks, such as finegrained image classification, where filters could learn to adapt to the object pose, or image deblurring, where filters can be tuned to adapt to the image structure.

## 6 Acknowledgements

This work was supported by FWO through the project G.0696.12N "Representations and algorithms for captation, visualization and manipulation of moving 3D objects, subjects and scenes", the EU FP7 project Europa2, the iMinds ICON project Footwork and bilateral Toyota project.

## Footnotes

*X. Jia and B. De Brabandere contributed equally to this work and are listed in alphabetical order.

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
