[Supplementary Material · dynamic-filter-networks-supplementary_material.pdf]

# Supplementary Material
# Dynamic Filter Networks

1 ## 1 Video Prediction

2 In this section, we add more results of the experiments on the moving MNIST and highway driving
3 datasets. We added an animated gif-file for every sequence we show here.

4 ## 1.1 Moving MNIST

Figure 1: Results on moving MNIST dataset with 2 moving digits. In the first row, the left half are
input sequence and the right half are ground truth of the future frames; in the second row, the right
half is the prediction of our model.

We visualize the dynamically generated filter for the first predicted frame.

Figure 2: Visualization of the dynamically generated filters. The first row are the previous frame, the second row are the visualization of filters, the third row are our prediction of the current frame and the fourth row are the ground truth.

6    We also run the model which is trained on 2 moving digits on out-of-domain data which has 1 moving digit or 3 moving digits. It shows the model has good generalization ability.

Figure 3: Results of out-of-domain input experiment with 1 moving digit and 3 moving digits.

 ## 1.2 Highway driving

Figure 4: Results on highway driving dataset.

## 2 Stereo Prediction

We make a video of stereo prediction results for the whole test sequence. In each frame, the first one is the left view image, the second one is the visualization of filters, the third one is our prediction of the right view, and the last one is the groundtruth.

## 3 Filter visualization

We can visualize the dynamically generated filters of the trained model in a flow-like manner. The filter's elements are non-negative and sum to one because of the use of softmax layer. To visualize a filter, we first calculate the horizontal and vertical distance of each of its elements to the center of the filter. Taking the sum of the x-axis shifts, weighted with the filter value at that location, we then obtain the overall x-axis shift caused by one filter. Similarly, we can get the overall y-axis shift. A filter can thus approximately be visualized as a 2-dimensional vector. For example, a 9x9 filter with a value of $1$ as its center top element can be seen as a vector with x-component $0$ and y-component $2$.

We then visualize the filters as an image where the hue and saturation of each pixel is set to respectively the orientation and magnitude of the filter at that location.