[Reviews · NeurIPS 2016]

Reviewer 1

Summary

This paper introduces a new framework termed "Dynamic Filter Networks", where the filters of a convolutional layer are not fixed but are predicted from a small subnetwork. The experiments contain a 1) sanity check (or a unit test) of the proposed approach, and networks that 2) take 10 frames of moving MNIST digits from [19] and predict the next 10 frames, 3) that predict 2 future frames in a highway driving application, and 4) that predict stereo views, posed as generating right view given left. In particular the experiments on moving MNIST show that the model can perform this task better than previous work and with many fewer parameters than previous work [18,19].

Qualitative Assessment

First, I'd like to say that this paper is very nicely written and was easy to read and understand. I also appreciated the thoroughness of experiments showing results in 1) unit test, 2) MNIST toy data, and 3) a slightly more real application. My major concern with this paper is with marketing the dynamic filter network as something entirely new. Certainly, the model predicts some parameters that happen to be used in a convolution, but that simply means that you have a slightly more complex mapping from inputs to outputs. The approach is still entirely parametric, you're merely tweaking the functional form of the mapping and allowing your network slightly more complex interactions. The authors devote a paragraph to this point on line 114, stating that: ``` we make a distinction between *model parameters* and *dynamically generated parameters*. Model parameters denote the layer parameters that are initialized in advance and only updated during training. They are the same for all samples. Dynamically generated parameters are sample-specific, and are produced on-the-fly without a need for initialization. ``` this is incorrect. The dynamic filter network is exactly identical to every other normal network: it still has a fixed set of parameters that are initialized in advance and updated during training. You're merely designing a more complex mapping. The definition of a parameter is not "the `weight` field in a convolutional layer". It is merely a number that changes during backprop. The authors spend an entire section 3.3 on drawing analogies to STNs, ResNets, etc. and I find these very far fetched and abstract comparisons. Finally, the experiments are nicely done and I appreciate Figure 7 that visualizes what the network learns. In summary, I think the authors make a big deal out of a simple idea, market it in a strange way, but ultimately do a nice job in execution: writing, experiments, and presenting the results. I think the experiments in this paper nicely highlight the strength of exactly this kind of model, where predicting the filters of a conv layer dynamically makes a lot of sense architecturally. Overall I like the paper and I'd be happy to see it accepted. Minor: - "Humans are good at predicting another view from related views". Are they? This is a poor way to start the intro of this paper. In general I'm not a fan of the first paragraph. - The authors have a strange interpretation of the Softmax function, claiming a few times that it helps "encourage elements to only have a few non-zero elements" (line 248). This is by no means the case. The network can choose to have high magnitude vectors where Softmax returns very peaky results, but it could also in principle return very diffuse vectors.

Confidence in this Review

3-Expert (read the paper in detail, know the area, quite certain of my opinion)


Reviewer 2

Summary

This paper proposes a model in which the filters of a convolutional layer (or spatially-locally-connected one) are produced as the output of a separate, trainable filter-generating network. This model is applied to a toy-problem (learning steerable filters) to demonstrate the workings, and to two more realistic problems to demonstrate the efficacy. Namely: video-prediction (with bouncing MNIST as well as a highway dataset), and stereo prediction. The model is shown to be effective in all cases.

Qualitative Assessment

This is a nice paper promoting a simple, but effective core idea — that parameters of a layer might be usefully generated based on the context of their input rather statically fixed for all examples. The experiments adequately demonstrate that the method can be effective, and good performance is show on synthetic and real-world tasks. To further demonstrate the power of the proposed method, it would be useful if the authors were able to show other examples in which multiple different (and successive) dynamic filter layers could be learned simultaneously in a deep net. The authors do a reasonable job at situating their proposal relative to other networks in the recent literature. However, one import reference/connection that seems to be lacking is a citation to Schmidhuber and colleague’s work on “Fast weights” (e.g. http://people.idsia.ch/~juergen/fastweights/ncfastweightsrev.html and ftp://ftp.idsia.ch/pub/juergen/icann2005gomez.pdf and related works). Some instantiations of this are in fact extremely similar to the current proposal. (e.g see figure 1 of the Gomez & Schmidhuber paper). The authors should discuss this. Neverthless, the work does have some novelty (e.g. the effective use of convolutions in the architecture). I believe that this work could have a reasonable impact on the field, given the way new modules that are effective have tended to proliferate in recent years. Particularly so, if it can be shown to be effective on larger problems.

Confidence in this Review

3-Expert (read the paper in detail, know the area, quite certain of my opinion)


Reviewer 3

Summary

The authors propose the idea of learning to predict the filters in a neural network based on the input and then apply these filters to the actual input. The idea is evaluated in video and stereo prediction tasks with a degree of success.

Qualitative Assessment

The proposed architectural idea is not particularly original, but if shown to work on multiple domains it could be very useful. What would make the idea more compelling though would be ablation studies that show that first predicting filters and then applying them really is a better method, then just learning filters directly. I find the choice of tasks to evaluate the proposed idea a bit limited. Does it work for plain classification? Other than that, the results on Moving MNIST seem to show a significant improvement. But it would be easier to assess the highway driving results if they were compared to some other baseline -- the few samples in the paper do not make these latter results that much more compelling.

Confidence in this Review

2-Confident (read it all; understood it all reasonably well)